# Jackal: A Real-World Execution-Based Benchmark Evaluating Large Language Models on Text-to-JQL Tasks

## Abstract

Enterprise teams rely on the Jira Query Language (JQL) to retrieve and filter issues from Jira. Yet, to our knowledge, there is no open, real-world, execution-based benchmark for mapping natural language queries to JQL. We introduce Jackal, a novel, large-scale text-to-JQL benchmark comprising 100,000 natural language (NL) requests paired with validated JQL queries and execution-based results on a live Jira instance with over 200,000 issues. To reflect real-world usage, each JQL query is associated with four types of user requests: (i) Long NL, (ii) Short NL, (iii) Semantically Similar, and (iv) Semantically Exact. We release Jackal, a corpus of 100,000 text-to-JQL pairs, together with an execution-based scoring toolkit, and a static snapshot of the evaluated Jira instance for reproducibility. We report text-to-JQL results on 23 Large Language Models (LLMs) spanning parameter sizes, open and closed source models, across execution accuracy, exact match, and canonical exact match. In this paper, we report results on Jackal-5K, a 5,000-pair subset of Jackal. On Jackal-5K, the best overall model (Gemini 2.5 Pro) achieves only 60.3% execution accuracy averaged equally across four user request types. Performance varies significantly across user request types: (i) Long NL (86.0%), (ii) Short NL (35.7%), (iii) Semantically Similar (22.7%), and (iv) Semantically Exact (99.3%). By benchmarking LLMs on their ability to produce correct and executable JQL queries, Jackal exposes the limitations of current state-of-the-art LLMs and sets a new, execution-based challenge for future research in Jira enterprise data.

## 1 Introduction

Retrieving structured information from enterprise systems is essential for triage, planning, governance, and reporting, yet end-users often struggle with the steep learning curve of formal query languages. Atlassian Jira, one of the most widely used issue tracking project management platforms, exemplifies this challenge. Its Jira Query Language (JQL), a domain-specific language (DSL), is expressive, supporting complex Boolean logic, custom fields, and temporal predicates, but its terse syntax and operator precedence present significant barriers for its 300,000+ users (Atlassian Support, 2025b). This motivates the development of LLM-powered natural language interfaces that map realistic, everyday user requests into executable JQL, saving users time and adding value to complex query logic.

Mapping natural language to executable programs is the central goal of semantic parsing (Liang, 2016). A common approach is learning from denotations, where models are supervised by execution results rather than gold programs (Guu et al., 2017). Progress in the text-to-DSL space has been driven largely by open, real-world Text-to-SQL benchmarks. WikiSQL established execution-based evaluation by checking whether predicted SQL yields the correct table answer (Zhong et al., 2017). Spider advanced the text-to-SQL field by testing cross-domain generalization with non-overlapping train and test schemas, where early exact match was only 12.4%, catalyzing research on compositional generalization (Yu et al., 2018). Later work showed that string match alone can miscount semantically correct programs and recommended execution-based scoring (Finegan-Dollak et al., 2018; Rajkumar et al., 2022). Complementary methods such as test-suite accuracy (Zhong et al., 2020) and constrained decoding (Scholak et al., 2021) further improved realistic assessment.

Natural language queries that mirror enterprise settings correlates with lower reported accuracy (Lei et al., 2025). Benchmarks that approximate production conditions, including richer schemas, multi-step interactions, and external context, consistently yield markedly lower scores than single-turn academic setups. Recent evaluations on enterprise-style Text-to-SQL (e.g., Spider 2.0) typically use multi-step or agentic procedures

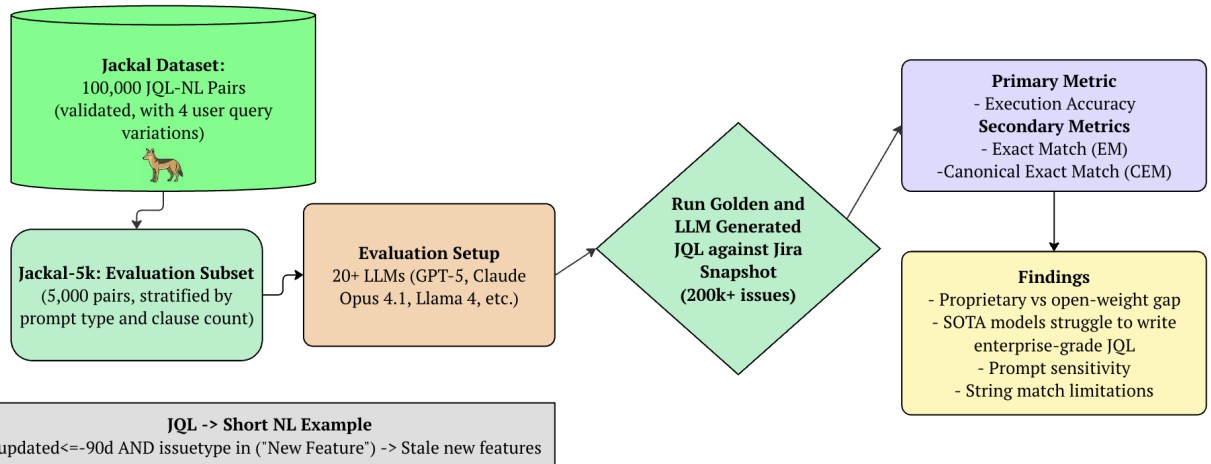

Figure 1: Overview of the Jackal benchmark. Jackal contains 100,000 real-world validated JQL–NL pairs with four user query variants. We evaluate 23 LLMs by running golden and generated queries against a live Jira instance (200K+ issues), using execution accuracy as the primary metric.

and report substantially lower success than classic single-turn datasets (e.g., Spider 1.0) or large but less interactive suites (e.g., BIRD), highlighting the evaluation gap between lab settings and production use (Lei et al., 2025; Yu et al., 2018; Li et al., 2023). JQL also presents distinct challenges relative to SQL, besides a fundamentally different syntax, it includes project- and instance-specific custom fields, permissions and visibility constraints, linked-issue traversals, and date functions, which further motivates execution-based evaluation.

Industry momentum also confirms the demand for natural language-to-JQL research. Atlassian Intelligence exposes natural language search that is translated into JQL in Jira, and several Marketplace applications advertise similar capabilities (Atlassian Support, 2025a; Clovity, 2025). However, these offerings are proprietary, with training data, prompts, and evaluation suites not publicly available, preventing open, independent research and execution-based assessment of LLMs. Only two public community datasets exist, Text2JQL and Text2JQL_v2 (Kulkarni, 2023a;b), but they are extremely limited in size (269 and 1,222 rows respectively), not real-world, do not mimic realistic user question complexity variations (short, long, semantics), do not provide an execution-based result of the Jira instance for evaluation, and are not reproducible.

To address the significant gap in text-to-JQL research, we introduce Jackal, the first open, large-scale, execution-based benchmark for mapping natural language queries to real-world JQL queries. Jackal contains 100,000 natural language user queries with validated JQL queries and verified execution results on a live Jira instance with more than 200,000 issues. To mimic diversity in user query phrasing, each natural language request is expressed via four NL query variants: (i) Long NL, an expanded sentence or short paragraph with context or reasoning; (ii) Short NL, a concise phrase that names the intent; (iii) Semantically Similar, a paraphrase that preserves intent without reusing JQL field names or values; and (iv) Semantically Exact, an easy literal translation that mirrors the JQL structure and order. Alongside the dataset, we release an evaluation toolkit that reports execution accuracy as the primary metric, which compares the live execution result predicted JQL match to the golden JQL execution result in Jackal, plus a static data dump of the evaluated instance to preserve reproducibility despite instance specificity. This paper evaluates 23 LLMs, both proprietary and open-source models, on Jackal-5K, a 5,000-pair subset of the 100,000 user-JQL pairs in Jackal. By filling the evaluation gap for natural language to JQL, we aim to accelerate open research on domain-specific language translation and lower the barrier for all Jira users to harness advanced search through LLMs.

In this paper we make three contributions:

- **Benchmark:** We release Jackal, the first real-world, open, execution-based dataset for natural language to JQL with four user query variants, validated queries on a live Jira instance, evaluation toolkit, and a reproducibility package.

- **Empirical Study:** We evaluate 23 language models on Jackal-5K across parameter scales and open/proprietary releases, reporting execution accuracy, exact match, and canonical exact match.
- **Analysis:** We provide a fine-grained error study, identifying frequent failure modes and suggesting directions such as grammar-constrained decoding and schema-aware retrieval.

## 2 RELATED WORKS

### 2.1 NATURAL LANGUAGE INTERFACE TO DATABASE AND EARLY SEMANTIC PARSING

Natural Language Interface to Database (NLIDB) and semantic parsing established the problem of mapping natural language to executable queries or programs. Early datasets include ATIS for spoken flight queries (Price, 1990; Hemphill et al., 1990) and GeoQuery for U.S. geography (Zelle & Mooney, 1996). Overnight showed rapid domain bootstrapping via grammar-induced logical forms (Wang et al., 2015). Surveys synthesize techniques and evaluation practices in modern NLIDB (Affolter et al., 2019). These works motivate domain-specific Text-to-DSL tasks and emphasize the importance of execution-based evaluation, setting the stage for large-scale Text-to-SQL benchmarks.

### 2.2 TEXT-TO-SQL BENCHMARKS AND CONVERSATIONAL EXTENSIONS

WikiSQL popularized large-scale, execution-based evaluation on single tables (Zhong et al., 2017). Spider shifted to cross-domain, multi-table, unseen-schema generalization with far lower early accuracy (Yu et al., 2018). Conversational and context-dependent parsing further stress realism via SParC and CoSQL (Yu et al., 2019b;a). Schema-aware encoders such as RAT-SQL improved Spider by better linking natural language mentions to schema elements (Wang et al., 2020). The BIRD benchmark scales content and database size, underscoring database-grounded, execution-based difficulty (Li et al., 2023). Most recently, Spider 2.0 reframes evaluation around enterprise workflows with very large schemas, multiple dialects, code and documentation context, and multi-step interactions, with markedly lower success than Spider 1.0 (Lei et al., 2025). This gap between lab settings and production use motivates domain-specific, execution-based benchmarks. Reported scores vary with metric and evaluation setup; Spider 2.0 under multi-step agentic evaluation is around 21.3%, Spider 1.0 under exact-match evaluation (non-agentic) is near 91.2%, and BIRD reports execution accuracy around 73.0%, so cross-benchmark figures are not directly comparable (Lei et al., 2025; Yu et al., 2018; Li et al., 2023).

### 2.3 EVALUATION METHODOLOGY BEYOND STRING MATCH

Exact string match can undercount or overcount semantically correct programs. The community converged on execution accuracy and variants like test-suite accuracy (Finegan-Dollak et al., 2018; Zhong et al., 2020). On the decoding side, execution-guided decoding prunes partial programs that fail at runtime (Wang et al., 2018), while constrained decoding such as PICARD enforces syntactic validity during generation (Scholak et al., 2021). Jackal follows this standard by centering execution-based scoring for Text-to-JQL, ensuring real-world validation against a live Jira instance, avoiding limitations in non-real-world evaluation methods.

### 2.4 TEXT-TO-DSL BEYOND SQL

Generalization to other formal languages reinforces our benchmark design choices. In Text-to-SPARQL for KBQA, LC-QuAD and LC-QuAD 2.0 pair natural language queries with SPARQL over DBpedia or Wikidata at larger scales (Trivedi et al., 2017; Dubey et al., 2019), and the QALD shared tasks standardize evaluation (Usbeck et al., 2017). For compositional generalization, CFQ offers a rigorous split construction used widely in semantic parsing (Keysers et al., 2020). In graph databases, recent Text2Cypher resources study Text-to-Cypher for Neo4j (Neo4j, 2024b;a). Program-as-state paradigms such as SMCalFlow in task-oriented dialogue demonstrate that execution-based supervision scales to other workflow-like languages (Andreas et al., 2020). Together, these lines show the breadth of Text-to-DSL mappings and the importance of execution-first evaluation.

### 2.5 TEXT-TO-JQL AND THE GAP TO OPEN, EXECUTION-BASED EVALUATION

JQL is a domain-specific language for querying issues in Jira. It supports Boolean logic, relational operators, field-based filtering, temporal predicates, and linked-issue traversals, but its terse syntax and operator

precedence can be a barrier for non-expert users. Unlike SQL, JQL operates over a structured yet non-relational data model in which issues, users, and projects are connected through fields, workflows, and permissions rather than normalized tables. Because each Jira instance may define custom fields, statuses, and workflows, and apply permission- and visibility-based access controls, the effective vocabulary and even the observable results of a query are instance- and user-specific. These characteristics make execution-based evaluation a necessity for Text-to-JQL (to measure practical utility) but also difficult to reproduce across environments.

Despite industry activity, open research in Text-to-JQL resources are significantly limited. Commercially, Atlassian provides natural language to JQL in Jira, and Marketplace apps promise similar features, but current evaluations are proprietary, closed-source, and not reproducible by the broader public (Atlassian Support, 2025a; Clovity, 2025). An enterprise-oriented study reports a small Text-to-JQL task with 218 prompts where GPT-4o achieved 54% exact match, but neither data nor scripts were released, which limits comparability and error analysis (Wang et al., 2025). Two public community datasets exist, Text2JQL (1,220 pairs) and Text2JQL_v2 (269 pairs), yet are extremely limited in size (269 and 1,222 rows respectively), not real-world, do not mimic realistic user question complexity variations (short, long, semantics), do not provide an execution-based result of the Jira instance for evaluation, and are not reproducible (Kulkarni, 2023a;b). This omission is particularly problematic for JQL, where instance-specific schemas, permissions, and linked-issue traversals can cause semantically valid predictions to be miscounted and non-executable strings to be scored as correct. JACKAL is the first benchmark to overcome these limitation by validating queries against a live real-world Jira instance of 200,000+ issues and releasing an execution-based evaluation toolkit tailored to JQL's idiosyncrasies, including custom fields, permissions and visibility, linked-issue traversals, and date functions.

## 3  THE JACKAL BENCHMARK

### 3.1  DATASET CONSTRUCTION

Jackal has a rigorous construction process, prioritizing real-world validation of JQL queries on a live jira instance, and user query variation to address diverse ways users ask queries in enterprise environments. Firstly, we generate candidate JQL queries by combining schema fields from a live Jira instance into compound filters with 2–5 clauses. Compound filters in Jira are common to "WHERE" conditions in SQL, in which more conditions lead to larger complexity. To avoid trivial or unrealistic queries for enterprise settings, we applied hand-crafted validity constraints, including rules that prevent contradictory field combinations (e.g., unassigned issues marked as fixed) or semantically incoherent pairs (e.g., low-priority bugs with blocker-level severity). Finally, to address a real-world limitation of live real-world benchmarks, each query was executed against a Jira instance with over 200,000 issues, and only those returning non-empty results were retained.

As stated previously, to increase diversity in queries, we sampled from multi-value groups, or clauses, for categorical fields such as issue type, priority, and resolution, and incorporated constraints to ensure broad coverage across date fields, text fields, and core schema elements. Larger clause count directly correlate to more complex JQL queries. This procedure yielded 100,000 valid, executable JQL queries. The overall dataset construction pipeline is illustrated in Figure 2.

#### 3.1.1  PROMPT VARIANTS

Once 100,000 validated JQL queries were programmatically generated, we used an LLM to generate the matching 100,000 user queries. To address a limitation of previous benchmarks, we design four natural language variations to mimic diverse usage in real-world enterprise settings:

- **Long NL:** Extended sentences with context or reasoning.
- **Short NL:** Concise phrases naming the intent.
- **Semantically Similar:** Paraphrases that preserve meaning without reusing JQL field names or values.
- **Semantically Exact:** Literal, field-by-field mappings that mirror JQL structure.

These variants were generated using a large language model with carefully designed prompts that enforce style consistency and ensure diversity across the four categories. See section 3.2 for examples of each user query variation.

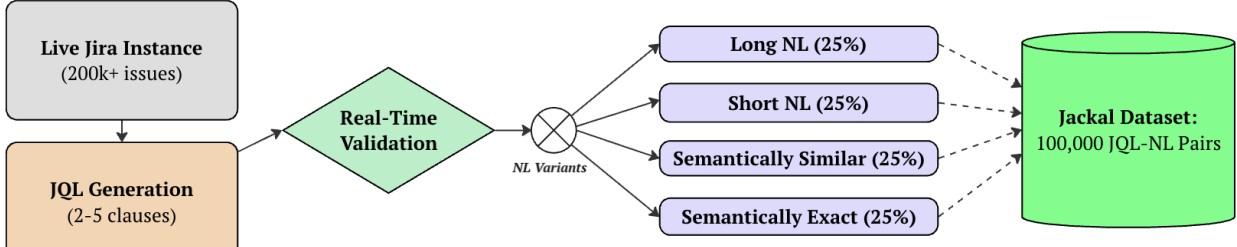

Figure 2: Jackal dataset construction pipeline. Candidate JQL queries are generated from a live Jira instance, validated through execution to filter out empty or invalid results, and then paired with four natural language variants before inclusion in the final 100K-pair dataset. In addition to execution filtering, we conducted human spot-checks to confirm the semantic fidelity of natural language variants and ensure dataset quality.

## 3.2 Statistics and Examples

Jackal contains 100,000 JQL-NL Query pairs, distributed evenly across clause counts (2-5) and 4 prompt variants. The average JQL query length is 3.5 clauses, with a vocabulary covering 15 Jira fields including project metadata, temporal filters, text search and custom fields. The average natural language request is 42.8 words for Long NL, 6.7 words for Short NL, 20.2 words for Semantically Similar, and 22.6 words for Semantically Exact. By combining validated executable queries with multiple variations of user queries that mimic real-world enterprise usage, Jackal provides a challenging, diverse, and reproducible benchmark for evaluating text-to-JQL systems.

Figure 2 illustrates the dataset construction pipeline for prompt variations. For instance, the JQL query: `updated<=-90d AND issuetype in ("New Feature")` is paired with the following variants:

- Long NL: *"I'm checking for any new feature requests that haven't been updated in the last 90 days, just to see if there are items that might have stalled or need attention."*
- Short NL: *"Stale new features"*
- Semantically Similar: *"Requests for new capabilities that haven't been changed in the last three months"*
- Semantically Exact: *'Updated is less than or equal to 90 days ago and issue type is New Feature"*

## 4 Evaluation Settings

### 4.1 Evaluation Dataset

We evaluate 23 open and closed-source LLM models against `Jackal-5K`, a 5,000-pair subset of the full benchmark. Jackal-5K is stratified by clause count and prompt variant to preserve balance across query complexity and linguistic styles. All results reported in this paper are computed on Jackal-5K, while the full 100,000 dataset supports large-scale training, analysis, and future research for the open community.

### 4.2 Metrics

We evaluate model outputs against gold JQL queries using three criteria:

- **Execution Accuracy:** Queries are executed against a snapshot of the Jira instance. A prediction is correct if it returns the same set of issue keys as the gold query. This is the primary measure of correctness, since multiple queries can be semantically equivalent even if their string forms differ.
- **Exact Match (EM):** Strict byte-level equality between generated and gold JQL after trimming whitespace.
- **Canonical Exact Match (CEM):** Equality after canonical normalization, which collapses whitespace and normalizes operators/keywords (e.g., AND, OR, IN). This captures formatting variation while preserving semantics.

| Model | Overall Average | Long NL | Short NL | Semantically Exact | Semantically Similar |
|---|---|---|---|---|---|
| Gemini 2.5 Pro | **0.603** | **0.860** | 0.345 | 0.991 | 0.215 |
| OpenAI o4-Mini | 0.595 | 0.848 | **0.357** | 0.974 | 0.200 |
| GPT-4o | 0.589 | 0.851 | 0.326 | 0.971 | 0.209 |
| Claude Sonnet 4 | 0.587 | 0.855 | 0.341 | 0.963 | 0.191 |
| OpenAI o3 | 0.586 | 0.856 | **0.357** | 0.923 | 0.206 |
| GPT-5 | 0.583 | 0.841 | 0.330 | 0.935 | **0.227** |
| Claude Opus 4.1 | 0.583 | 0.843 | 0.344 | 0.926 | 0.219 |
| Gemini 2.5 Flash | 0.581 | 0.823 | 0.311 | **0.993** | 0.197 |
| GPT-4.1 | 0.580 | 0.846 | 0.334 | 0.940 | 0.199 |
| GPT-5 Mini | 0.578 | 0.829 | 0.315 | 0.969 | 0.198 |
| Claude Opus 4 | 0.576 | 0.838 | 0.335 | 0.915 | 0.215 |
| OpenAI o3-Mini | 0.576 | 0.841 | 0.330 | 0.957 | 0.175 |
| Claude Sonnet 3.7 | 0.572 | 0.814 | 0.299 | 0.986 | 0.190 |
| Llama 4 Maverick (17B) | 0.563 | 0.798 | 0.310 | 0.953 | 0.193 |
| GPT-4.1 Mini | 0.562 | 0.834 | 0.285 | 0.936 | 0.193 |
| Llama 3.3 (70B) | 0.545 | 0.799 | 0.236 | 0.981 | 0.163 |
| GPT-4o Mini | 0.531 | 0.776 | 0.219 | 0.965 | 0.164 |
| Nova Pro | 0.489 | 0.714 | 0.215 | 0.867 | 0.160 |
| Gemini 2.5 Flash-Lite | 0.465 | 0.625 | 0.129 | 0.935 | 0.170 |
| Llama 3 (70B) | 0.450 | 0.648 | 0.177 | 0.843 | 0.130 |
| Llama 3 (8B) | 0.301 | 0.433 | 0.086 | 0.627 | 0.057 |
| Mistral 7B | 0.215 | 0.224 | 0.026 | 0.580 | 0.030 |
| Mixtral 8x7B | 0.191 | 0.277 | 0.055 | 0.401 | 0.031 |
| *LLM Model Average* | *0.517* | *0.742* | *0.264* | *0.893* | *0.171* |

Table 1: LLM execution accuracy on Jackal-5K (5,000 text-to-JQL pairs), sorted by *Overall Average.* The first column reports each model's overall average (equal dataset weight, 0.25 each) across four request types; subsequent columns show Long Natural Language (NL), Short Natural Language (NL), Semantically Exact, and Semantically Similar, with the bottom row giving macro-averages across models.

## 4.3 EVALUATION SETUP

For each model and prompt variant, we pair the generated JQL with its gold query and evaluate along these three axes. Both predicted and gold queries are executed against the same Jira snapshot via the `/rest/api/2/search` endpoint. We stratify analysis by prompt type (Long NL, Short NL, Semantically Similar, Semantically Exact) and also report aggregate performance.

Results are presented per model, per prompt type, and overall. We show execution accuracy breakdowns (`match`, `mismatch`), mean EM and CEM by model and prompt type, and comparative visualizations highlighting variation across LLMs. All reported results use **Jackal-5K**, ensuring reproducibility and comparability, while the full 100K dataset supports large-scale training.

## 5 EXPERIMENTS

### 5.1 SETUP

We evaluate 23 proprietary and open-weight LLMs on **Jackal-5K**, the standardized 5,000-pair evaluation subset of Jackal. Each model is tested across all four user query variants (Long NL, Short NL, Semantically Similar, Exact Match). For each prediction, we compute (i) execution accuracy, (ii) exact match (EM), and (iii) canonical exact match (CEM), as described in Section 4.1. All queries are executed against the same snapshot of the Jira instance to ensure reproducibility.

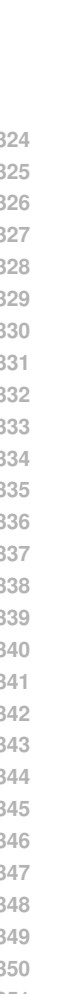
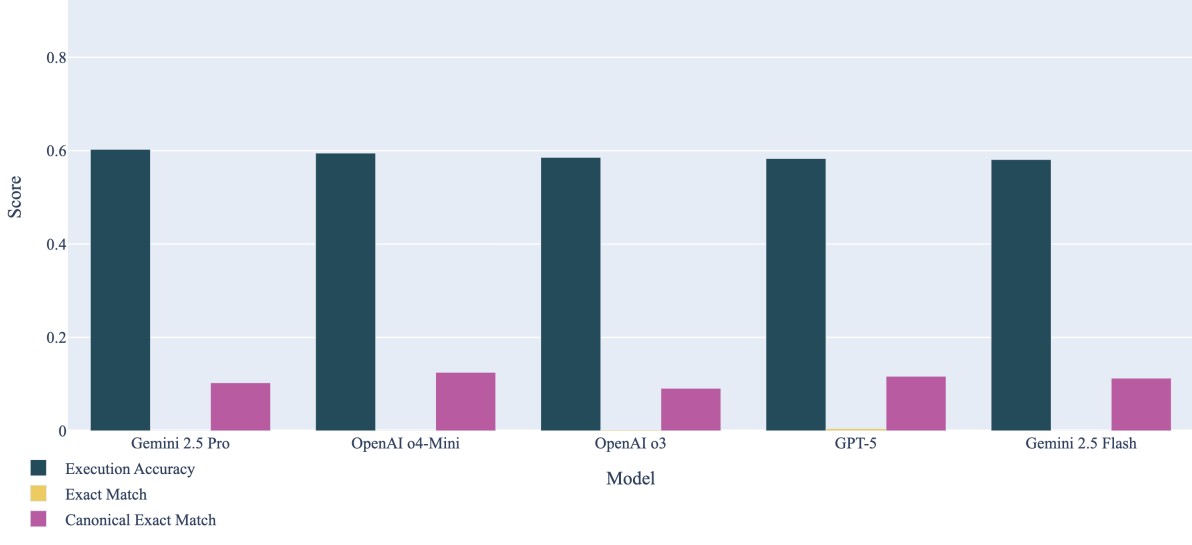

Figure 3: Execution Accuracy, Exact Match (EM), and Canonical Exact Match (CEM) on Jackal-5K for the five leading LLMs by overall average user query variant performance. While top models achieve strong execution accuracy, EM and CEM remain near zero, highlighting the insufficiency of string-level metrics.

## 5.2 Overall Results

Table 1 summarizes model performance. Across all models, the average execution accuracy is 0.517, with large variation across user query variations: 0.742 for Long NL, 0.264 for Short NL, 0.893 for Semantically Exact, and 0.171 for Semantically Similar. The best-performing model is Gemini 2.5 Pro, reaching 0.603 overall execution accuracy, with particularly strong results on Long NL (0.860) and Semantically Exact (0.991). Proprietary models such as GPT-4o (0.589 execution accuracy) and Claude-4 (0.588 execution accuracy) perform consistently well, though their accuracy on Semantically Similar variation remains below 0.21. In contrast, open-weight models lag significantly: Llama 3 (70B) averages 0.450 execution accuracy, while smaller variants such as Llama 3 (8B) and Mistral-7B achieve only 0.301 and 0.215 execution accuracy, respectively. These results confirm a persistent performance gap between frontier proprietary and open-weight models.

## 5.3 Exact and Canonical Match

Figures 3 reports exact match (EM) and canonical exact match (CEM). As expected, raw EM is nearly zero across all models, with a global average of 0.0008. Even the highest EM values (OpenAI o3-mini at 0.006 and Claude Opus 4.1 at 0.005) are negligible compared to execution accuracy. Canonical normalization provides modest improvements: the average CEM is 0.099, with top models (e.g., Llama 4 Maverick 17B and GPT-4o Mini) reaching 0.136. Still, these values remain far below execution accuracy, reinforcing that string-level metrics fail to capture semantic correctness in text-to-JQL tasks.

## 5.4 Findings

Our evaluation highlights three key observations:

- **User Query Variation:** Performance varies drastically across user query variations types. While models excel on Semantically Exact variation (average execution accuracy 0.893), accuracy collapses for Short NL (0.264) and Semantically Similar (0.171), underscoring the challenge of linguistic variability.
- **Proprietary vs open-weight gap:** Proprietary models consistently outperform open-weight ones. For example, Gemini 2.5 Pro (0.603 execution accuracy) and GPT-4o (0.589 execution accuracy) surpass Llama 4 Maverick (17B) at 0.563 execution accuracy, while small open-weight models fall below 0.31 execution accuracy.

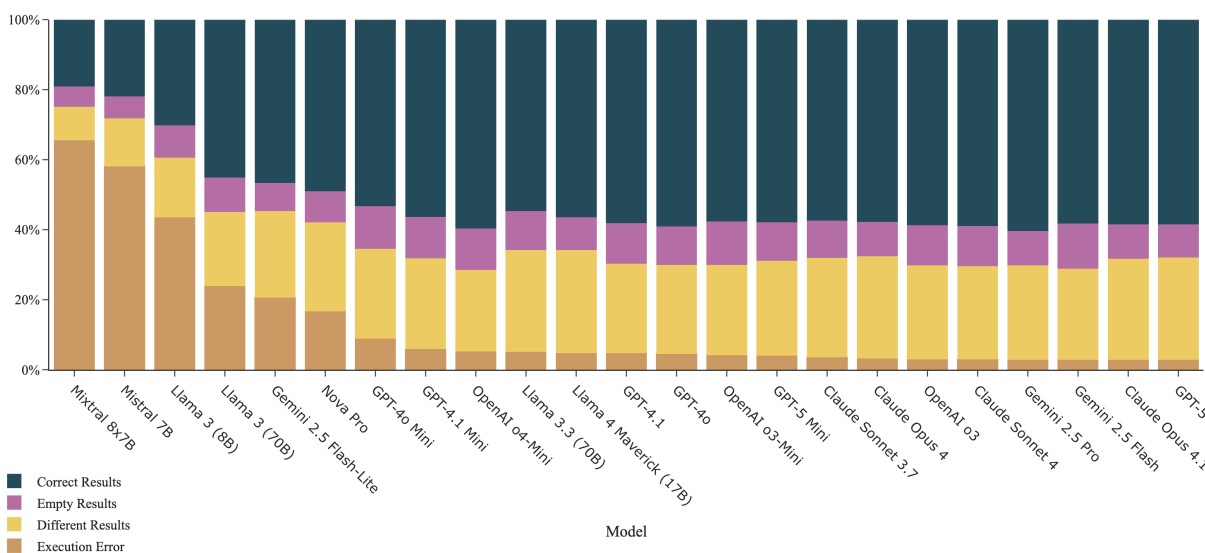

Figure 4: Error analysis on Jackal-5K across 23 LLMs. Each bar shows the proportion of predictions falling into four categories: Correct Results (generated JQL returning the same issue set as the golden JQL), Empty Results (generated JQL returning no issues), Different Results (generated JQL returning an issue set different than the golden JQL), and Execution Errors (invalid JQL that fails to run). s Smaller open-weight models (e.g., Mistral 8x7B, Mistral 7B, Llama 3 8B) are dominated by execution errors, whereas stronger proprietary models are mostly limited by different or empty results.

- **String match limitations:** EM and CEM remain near zero, providing little evaluative value compared to execution accuracy. This validates the critical limitation of non-real-world benchmarks that do not use execution-based evaluation as the primary metric. In Jackal, we solve this limitation by using execution-based evaluation.

## 6 ANALYSIS

### 6.1 USER QUERY VARIATION SENSITIVITY

Performance varies sharply across user query variations. Models achieve their highest scores on the Semantically Exact variation, with an average execution accuracy of 0.893. In contrast, accuracy falls to 0.742 for Long NL, 0.264 for Short NL, and only 0.171 for Semantically Similar variation. These results indicate that while models can reliably handle literal, field-by-field translations into JQL, they struggle with under-specified (Short NL) or paraphrased (Semantically Similar) inputs. This gap underscores the brittleness of current approaches to linguistic variability and paraphrase resolution.

### 6.2 PROPRIETARY VS. OPEN-WEIGHT MODELS

Frontier proprietary models consistently outperform open-weight alternatives. Gemini 2.5 Pro leads overall with 0.603 execution accuracy, followed by OpenAI o4-mini (0.595), GPT-4o (0.580), and Claude Sonnet 4 (0.587). By contrast, the strongest open-weight model, Llama 4 Maverick (17B), reaches 0.563 execution average, while smaller open models such as Llama 3 (8B) and Mistral-7B drop to 0.301 and 0.218 execution accuracy, respectively. Even the best proprietary models, however, fall below 0.23 execution accuracy on Semantically Similar prompts, showing that robustness to natural variation remains unresolved.

### 6.3 STRING MATCH LIMITATIONS

Exact string match (EM) and canonical exact match (CEM) provide little signal compared to execution accuracy. The global average EM is only 0.0008, effectively zero across models, with the highest single score being 0.005 (Claude Opus 4.1). Canonical normalization recovers modest alignment, raising the average

CEM to 0.099, with top scores of 0.136 (Llama 4 Maverick 17B and GPT-4o Mini). These values remain far below execution accuracy (average 0.517), demonstrating that string-based metrics systematically undercount semantically correct predictions. Execution-based evaluation is therefore essential for realistic assessment.

## 6.4 Error Categories

To better understand where models fail, we analyzed predictions across user query variants and grouped them into the same four categories reported in Figure 4:

- **Correct Results:** Generated JQL queries whose returned issue set exactly matches the gold query. Frontier proprietary models achieve the highest rates of correct results.
- **Empty Results:** Generated JQL queries that return no issues. These typically arise from over-specified constraints or misaligned filters, and are especially common on Short NL prompts.
- **Different Results:** Generated JQL queries that execute but return a different set of issues than the gold query. These reflect semantic mismatches in field or value selection, often seen in Semantically Similar prompts.
- **Execution Errors:** Invalid JQL queries that fail to run due to syntax or schema errors. Smaller open-weight models (e.g., Mistral 8x7B, Mistral 7B, Llama 3 8B) are dominated by execution errors, while proprietary models largely avoid them.

Overall, the error distribution highlights distinct failure modes: weaker open-weight models struggle primarily with execution errors, while stronger proprietary models are mostly limited by producing queries that return empty or mismatched result sets.

Together, these categories illustrate that both surface-level errors and deeper semantic mismatches contribute to the performance gap across prompt types.

## 7 Conclusion

We introduce Jackal, the first open, real-world, execution-based benchmark for mapping natural language to Jira Query Language (JQL), directly addressing the limitations of prior work: open-source, small or fictional dataset, string-match evaluations in place of live execution-based, a lack of realistic user query phrasing, and no reproducibility. Jackal consists of 100,000 natural language (NL) user requests with validated JQL and recorded execution results on a live Jira instance with 200,000+ issues. Each user query is varied across four diverse methods: (i) Long Natural Language Queries (Long NL), (ii) Short Natural Language Queries (Short NL), (iii) Semantically Similar, and (iv) Semantically Exact. We release the Jackal dataset, an execution-based evaluation toolkit, and a static snapshot of the evaluated instance to ensure reproducible, open evaluation for research. Furthermore, we evaluate 23 LLMs on Jackal-5K, a 5,000 subset of Jackal 100,000, and report the highest overall model (Gemini 2.5 Pro) attains only 60.3% execution accuracy when averaging equally across request types. User query variant contributes to a significant disparity in results, with 86.0% (Long NL), 35.7% (Short NL), 22.7% (Semantically Similar), and 99.3% (Semantically Exact). By centering execution accuracy rather than string match and grounding evaluation in real-world Jira data with reproducible artifacts, Jackal exposes the limitations of current state-of-the-art LLM models and establishes a rigorous, scalable foundation for future research on text-to-JQL.

## Use of LLMs

Large Language Models (LLMs) were not used significantly in writing the paper. LLMs were used to aid or polish writing. Details of their use for the dataset are described in the paper.

## Ethics Statement

All authors have read and adhere to the ICLR Code of Ethics (`https://iclr.cc/public/CodeOfEthics`). This research does not involve human subjects, sensitive personal data, or interventions, and no datasets released contain confidential or proprietary information. The benchmark and evaluation procedures are designed to promote reproducibility and open research. The use of LLMs for text generation was limited to

writing support and dataset paraphrasing, and human validation was conducted to ensure fidelity. We believe there are no foreseeable harms arising from this work beyond the ordinary risks associated with releasing benchmarks and evaluation code in machine learning research.

## Reproducibility Statement

We have taken several steps to ensure reproducibility. The benchmark dataset (Jackal) and evaluation toolkit will be released with full documentation, including the schema generation process, execution validation pipeline, and natural language variant creation procedure (see Section 5). A reproducible subset (Jackal-5K) is provided for evaluation. To address the limitation of the live Jira instance, we release a static data dump of the 200,000+ instances that can be used in future evaluations. A reproducible subset (Jackal-5K) is provided for evaluation, along with execution results against a static snapshot of the Jira instance. All model outputs reported in this paper were evaluated against this fixed benchmark. Additional details, including dataset construction steps, examples, and evaluation code, are provided in the appendix and supplementary materials.

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

## A   Full JQL Query Examples

=

Table 2: Examples of JQL queries with increasing clause complexity, each paired with four natural-language request types.

| No. of Clauses | JQL | Long NL | Semantically Exact | Semantically Similar | Short NL |
|---|---|---|---|---|---|
| 2 | `updated <= -90d AND issuetype in ("Epic")` | I'm checking for epics that haven't been updated in the last 90 days, so we can identify any long-standing items that might need attention or could potentially be closed. | Updated is less than or equal to 90 days ago, and issue type is Epic | Large-scale tasks that haven't been changed in the last three months | Epics inactive for 90+ days |
| 3 | `created >= -4w AND assignee is EMPTY AND issuetype in ("User Story")` | I'm checking for any user stories that have been created in the last four weeks but haven't been assigned to anyone yet, so we can make sure nothing important is slipping through the cracks. | Created is within the last 4 weeks, assignee is empty, and issue type is User Story | Stories added in the last month that haven't been assigned to anyone yet | Unassigned user stories created in last 4 weeks |
| 4 | `updated >= "2025-01-01" AND issuetype in ("Bug") AND priority is not EMPTY AND description "crash"` | I'm searching for bug reports that have been updated since the start of 2025, specifically those that mention a crash in their description and have a priority set. This helps me focus on recent and prioritized crash-related issues that might need urgent attention. | Updated is on or after 2025-01-01, issue type is Bug, priority is not empty, and description contains crash | Problems flagged as bugs, with a set urgency level, mentioning crashes, and modified after the start of 2025 | Bugs with priority and crash in description updated since 2025 |
| 5 | `updated <= "2025-01-01" AND description "error" AND affectedVersion is not EMPTY AND resolution is EMPTY AND issuetype in ("Epic", "User Story", "Task", "Sub-task")` | I'm looking for any Epics, User Stories, Tasks, or Sub-tasks that mention 'error' in their description, have at least one affected version specified, haven't been resolved yet, and were last updated on or before January 1st, 2025. This helps me identify older, unresolved issues related to errors that might need attention or follow-up. | Updated is on or before 2025-01-01, description contains error, affected version is not empty, resolution is empty, and issue type is Epic, User Story, Task, or Sub-task | Items that mention an error, haven't been resolved yet, are linked to a specific version, and were last changed before January 2025, covering all major and minor work categories. | Unresolved issues with error in description |

*Note: Each JQL query is paired with four natural-language request types — Long NL (extended description with context), Semantically Exact (literal mapping to JQL), Semantically Similar (paraphrased intent), and Short NL (concise label).*

## B   User Query Variation Prompts

This appendix provides the four prompt templates used to generate natural language queries from JQL. Each template enforces a distinct variation: Long NL, Short NL, Semantically Similar, and Semantically Exact.

## B.1 Long Natural Language (Long NL)

Task: Convert a JQL query into a longer natural language sentence or paragraph.
Include context or reasoning that someone might give when discussing the query aloud.
Vary the style, making some responses conversational or explanatory.

```
Examples:
1. JQL: project = QTBUG AND issuetype = Bug AND status = "Open"
   NL: I'm reviewing all open bugs in the QTBUG project so we can track unresolved
       issues before the next sprint.

2. JQL: created >= -5d AND project = PYSIDE
   NL: I want to look at issues reported in the last 5 days in the PYSIDE project
       to see what's newly come in.
```

Given this JQL: {jql}

```
OUTPUT FORMAT:
Only respond with the natural language.
Example: "I'm reviewing all open bugs in the QTBUG project so we can track unresolved
issues before the next sprint."
Do not include any additional text or explanations.
```

## B.2 Short Natural Language (Short NL)

Task: Convert a JQL query into a concise natural language phrase, just a few words.
Prefer minimal and direct expressions.

```
Examples:
1. JQL: project = QDS AND priority = "P0: Blocker"
   NL: QDS blockers

2. JQL: resolution = Duplicate
   NL: Duplicate issues
```

Given this JQL: {jql}

```
OUTPUT FORMAT:
Only respond with the natural language.
Example: "Duplicate issues"
Do not include any additional text or explanations.
```

## B.3 Semantically Similar

Task: Convert the following JQL query into a natural language sentence that expresses
the same intent, but uses different wording. Do not directly reuse JQL field names or
values. Instead, rephrase using synonyms, conversational language, or implied meaning.
Be creative, but maintain accuracy.

```
Examples:
1. JQL: status = "Open"
   NL: Tickets that are still in progress

2. JQL: resolution = Duplicate
   NL: Issues already reported before
```

Given this JQL: {jql}

```
OUTPUT FORMAT:
```

```
Only respond with the natural language.
Example: "Issues already reported before"
Do not include any additional text or explanations.
```

### B.4 Semantically Exact

```
Task: Translate a JQL query into a natural language sentence that mirrors the JQL
structure and wording as closely as possible. Do not paraphrase or add unnecessary
context. Use literal conversions of fields and values, preserving the order and logic.

Examples:
1. JQL: project = QTBUG AND issuetype = Bug AND status = "Open"
   NL: Project is QTBUG, issue type is Bug, and status is Open

2. JQL: priority = "P1: Critical"
   NL: Priority is P1: Critical

Given this JQL: {jql}

OUTPUT FORMAT:
Only respond with the natural language.
Example: "Project is QTBUG, issue type is Bug, and status is Open"
Do not include any additional text or explanations.
```

## C Prompt Template for JQL Generation

This prompt template was used to generate JQL queries from natural language inputs with schema grounding. The schema provides the allowed fields ('jqlName' keys), ensuring that model outputs remain valid within the target Jira instance.

```
PROMPT_TEMPLATE_with_schema = """
Given the following natural language description of a Jira search query,
generate the corresponding valid JQL (Jira Query Language) query.
Use only fields from the SCHEMA below ('jqlName' as the field key).

SCHEMA:
{schema}

Natural language:
"{natural_language}"

Output only valid JSON in this format:
{
   "jql": "<the generated JQL query>"
}
"""
```

## D Summarized JQL Schema

This appendix provides a summarized version of the Jira schema used to ground JQL generation. It lists the major fields, their JQL keys, supported operator types, and representative value categories. All project-specific or proprietary values have been omitted.

- **Issue Type (issuetype)** – categorical; operators: `=, !=, IN, NOT IN`; example values: Bug, Epic, User Story, Task, Sub-task.
- **Project (project)** – categorical; operators: `=, !=, IN, NOT IN`; values omitted for confidentiality.

- **Components (component)** – categorical; operators: `=, !=, IN, NOT IN, IS EMPTY, IS NOT EMPTY`; large enum set (not listed).

- **Platforms (custom field)** – categorical; operators: `=, !=, IN, NOT IN, IS EMPTY, IS NOT EMPTY`; example values: Windows, Linux, macOS, Android, iOS.

- **Labels (labels)** – categorical; operators: `=, !=, IN, NOT IN, IS EMPTY, IS NOT EMPTY`; values drawn from natural language mentions.

- **Fix Version/s (fixVersion)** – categorical; operators: same as Labels; values omitted due to size.

- **Affects Version/s (affectedVersion)** – categorical; operators: same as Labels; values omitted due to size.

- **Resolution (resolution)** – categorical; operators: `=, !=, IN, NOT IN, IS EMPTY, IS NOT EMPTY`; example values: Fixed, Duplicate, Invalid, Won't Do.

- **Priority (priority)** – categorical; operators: `=, !=, IN, NOT IN`; example values: Blocker, Critical, Important, Low.

- **Summary (summary)** – text search; operators: `~, !~`.

- **Description (description)** – text search; operators: `~, !~`.

- **Assignee (assignee)** – categorical/special; operators: `=, !=, IS EMPTY, IS NOT EMPTY`.

- **Created (created)** – date; operators: `>=, <=, >, <, =`; supports absolute dates (YYYY-MM-DD), relative dates (e.g., -5d, -4w), and functions (e.g., startOfMonth()).

- **Updated (updated)** – date; same operators and hints as Created.

- **Resolved (resolutiondate)** – date; same operators and hints as Created.

**Aliases.** Common natural language mentions were normalized to schema keys. For example: "issue type" → `issuetype`, "fix version/s" → `fixVersion`, "affects version/s" → `affectedVersion`, "components" → `component`, "platforms" → custom field.

