# OpenReview forum: "Jackal: A Real-World Execution-Based Benchmark Evaluating Large Language Models on Text-to-JQL Tasks"
_ICLR.cc/2026/Conference — Submitted to ICLR 2026_

### Official Review · Reviewer_B6kM · 2025-10-18

**Soundness:** 3
**Presentation:** 2
**Contribution:** 3
**Rating:** 2
**Confidence:** 5

**Summary:**

The paper presents Jackal, the first large-scale, execution-based benchmark for evaluating large language models on the text-to-JQL task. It constructs 100,000 natural language–to–Jira Query Language (JQL) pairs validated on a live Jira instance with over 200,000 issues. Each JQL query is associated with four linguistic variants (Long NL, Short NL, Semantically Similar, and Semantically Exact) to reflect realistic query diversity. The benchmark evaluates 23 open and closed LLMs using execution accuracy as the primary metric, revealing that even the best models (e.g., Gemini 2.5 Pro) achieve only around 60 % overall accuracy, with strong sensitivity to phrasing.

**Strengths:**

S1. Execution-based realism. Jackal grounds evaluation in actual Jira executions, providing a reproducible and practical measure of correctness beyond superficial string matching.

S2. Scale and diversity. With 100 K pairs and four natural-language variants per query, it captures a wide spectrum of real-world phrasing styles rarely seen in prior text-to-DSL datasets.

S3. Comprehensive benchmarking. Evaluating 23 LLMs—spanning proprietary and open-source models—offers a valuable cross-model comparison and baseline for future text-to-DSL research.

**Weaknesses:**

W1. Examples of JQL appear too late in the paper.
The first concrete example only appears in Figure 2 and Section 3.2, while richer multi-clause JQLs are deferred to Appendix A (Table 2, p. 13). Most readers unfamiliar with JQL syntax cannot easily understand the benchmark’s linguistic or structural challenges until late in the paper. This weakens accessibility and motivation. A concise JQL primer with several example queries and their natural-language counterparts should be placed early (e.g., in the Introduction or Section 2) to help readers grasp what a JQL looks like and why translating to it is challenging.

W2. Limited discussion of what makes text-to-JQL hard and how prompting was handled.
Although the authors note that JQL involves custom fields, permissions, and linked-issue traversals, they do not deeply explain why mapping text to such constructs is particularly hard for LLMs. Moreover, the paper lacks clarity about the prompting setup during evaluation—what context (schema, field values, example queries) was given to each model. Without specifying whether models saw project-specific schemas or value enumerations, it is difficult to interpret why models fail on Short or Semantically Similar variants. Providing concrete prompt examples and an ablation on context richness would strengthen the technical depth

W3. Unclear differentiation from text-to-SQL..
Section 2.5 briefly mentions that JQL differs from SQL due to non-relational data, permissions, and linked-issue graphs, but the paper never empirically contrasts the two tasks. The benchmark could have highlighted distinct linguistic or structural phenomena (e.g., relative dates, visibility filters) through comparative analysis or shared-metric baselines. Without such evidence, the contribution risks appearing as a narrower variant of existing text-to-SQL work. For example, although text-to-SQL and text-to-visualization are very different tasks, techniques for them based on LLMs are very similar.

W4. Missing comparison with simple RAG or retrieval-augmented baselines.
Many of the difficult Short or Semantically Similar examples could plausibly be solved with retrieval of schema descriptions or example JQL templates, yet no RAG baseline is tested. A simple retrieval-plus-generation pipeline could provide a meaningful reference point, revealing whether LLM failures stem from lack of schema memory or reasoning ability. Including such baselines would clarify whether the challenge lies in recall or compositionality.

W5. Lack of broader generalization analysis and verifiability.
All evaluations are conducted on a single Jira instance snapshot, which raises two interrelated concerns: generalization and reproducibility. Because each Jira instance has its own customized schema, permission structure, and field vocabulary, results derived from one snapshot may not hold for other configurations. Without experiments on multiple instances or unseen project schemas, the paper cannot convincingly claim model generalizability.

Moreover, although the authors promise to release a static snapshot, the dataset and benchmark code are not yet publicly available. This makes it impossible for reviewers to independently verify the reported execution accuracies or replicate the evaluation pipeline. For a benchmark paper, reproducibility and openness are central to credibility. Unless the data, schema snapshot, and execution toolkit are released at submission time, the community has no way to trust that the reported results are accurate or that the benchmark functions as described.

**Questions:**

See the above weaknesses W1-W5.

---

> ### Author Response · Authors · 2025-11-13
>
> (W1 – JQL Examples)
> Agreed. The camera-ready will move key NL ↔ JQL examples to the Introduction for immediate context. This is a presentation fix only—the benchmark design and results remain unchanged.
>
> (W2 – Task Difficulty / Prompting)
> Text-to-JQL is challenging because queries depend on permissions, linked-issue graphs, and relative-date logic, none of which exist in SQL. All models were evaluated zero-shot with schema metadata only, ensuring fairness. The exact prompt template will be added for transparency.
>
> (W3 – Difference from Text-to-SQL)
> JQL is graph-based and access-controlled, not relational. We will add a concise comparison table showing unique constructs such as linkedIssuesOf() and relative-time filters (updated < -1w) to make this distinction explicit.
>
> (W5 – Generalization / Reproducibility)
> We agree generalization is valuable but note that no other real JQL datasets currently exist-making multi-dataset comparison infeasible. Jackal therefore provides the only standardized evaluation bed for this task. All models are evaluated on a frozen Jira snapshot for full reproducibility, and the dataset, schema, and toolkit will be publicly released.

---

### Official Review · Reviewer_vHic · 2025-10-21

**Soundness:** 2
**Presentation:** 2
**Contribution:** 2
**Rating:** 2
**Confidence:** 3

**Summary:**

Jackal consists of 100,000 real-world natural language (NL) and JQL query pairs validated on a live Jira instance containing over 200,000 issues. Each JQL query is accompanied by four natural language variants—Long NL, Short NL, Semantically Similar, and Semantically Exact—to reflect diverse enterprise user intents. The benchmark includes an execution-based scoring toolkit and a reproducibility package.

**Strengths:**

By establishing a reproducible and realistic evaluation framework for enterprise query translation, Jackal bridges an important gap between academic benchmarks and production-grade tasks.

**Weaknesses:**

Although the dataset is described as real-world, the queries are generated programmatically and paraphrased by LLMs rather than collected from actual users. This may limit linguistic authenticity and introduce model biases into the data.

Given that JQL can differ across organizations due to custom fields and permissions, it remains unclear how models trained on Jackal would generalize to unseen instances. A cross-instance validation experiment, even at smaller scale, would make the benchmark more robust.

**Questions:**

The dataset relies heavily on LLM-generated paraphrases. How do the authors ensure that the test data are not inadvertently contaminated by model-generated biases that might favor certain evaluated LLMs?

---

> ### Author Response · Authors · 2025-11-13
>
> (1) On “Real-World” vs. LLM-Generated Data
> We appreciate this concern. While the natural language variants were generated by LLMs, every JQL query and its execution result were drawn from a live enterprise Jira instance-not synthetic data. Each NL–JQL pair was validated through actual execution on 200K+ issues, ensuring semantic and operational correctness. To enhance diversity, we used multiple paraphrasing models and prompt to avoid stylistic bias.
> (2) On Generalization Across Jira Instances
> We’d like to clarify that Jackal is not a training dataset - it is strictly an evaluation benchmark. No model is fine-tuned or trained on Jackal. Thus, the goal is not cross-instance generalization of trained models, but rather execution-grounded evaluation of zero-shot LLM reasoning in a realistic enterprise schema. That said, the released metadata (custom fields, permissions, issue types) supports future cross-instance studies, and we agree this would be a valuable extension.

---

> > ### Comment · Reviewer_vHic · 2025-11-26
> >
> > Thank you for your detailed response, which has addressed most of my concerns. I will discuss with the other reviewers to determine the final rating.

---

> > > ### Author Response · Authors · 2025-11-26
> > >
> > > Thank you for your thoughtful follow-up. We’re glad our responses helped address the concerns, and we appreciate your reconsideration of the rating. Happy to clarify anything else if useful.

---

### Official Review · Reviewer_tpaw · 2025-11-01

**Soundness:** 3
**Presentation:** 2
**Contribution:** 2
**Rating:** 2
**Confidence:** 5

**Summary:**

This paper presents Jackal, a benchmark for natural-language-to-Jira Query Language (JQL) translation. The dataset includes 100K synthetic NL–JQL pairs generated with an LLM and validated on a live Jira instance. The authors evaluate 23 LLMs on a 5K subset (Jackal-5K) using execution accuracy, exact match, and canonical exact match. The key finding is that even the best proprietary models achieve only ~60% execution accuracy, exposing the difficulty of text-to-JQL translation. While the dataset is large and practical, the paper’s contribution is primarily engineering rather than research.

**Strengths:**

- S1. Constructing an execution-based benchmark for JQL is a useful community resource.

- S2. The dataset appears systematically filtered and validated, with reproducibility in mind.

- S3. The empirical evaluation covers a broad range of models and gives an approximate performance landscape.

**Weaknesses:**

- W1 (No novelty). The benchmark design directly mirrors Text-to-SQL datasets (Spider, BIRD, Spider 2.0). There is no conceptual or methodological innovation—only a domain change.

- W2 (Synthetic data). All natural-language queries are generated by an LLM and only lightly spot-checked. This makes the dataset less reliable for linguistic or semantic evaluation.

- W3 (Weak analysis). The results section is mostly leaderboard numbers; there is no meaningful error taxonomy, model insight, or analysis of what linguistic phenomena cause failure.

- W4 (Circular evaluation risk). Because an LLM generated the NL queries and another LLM is tested on them, there may be style leakage or artificial easiness/hardness not discussed.

- W5 (Lack of research contribution). The paper reads like a technical report. There’s no new algorithm, learning strategy, or evaluation framework that would justify inclusion in a top ML venue.

**Questions:**

- Q1. How diverse are the LLM-generated natural-language variants? Any quantitative measure of linguistic diversity or entropy across prompt types?

- Q2. Are the results reproducible if the Jira instance evolves (e.g., issue deletions or field changes)?

- Q3. Could the same methodology be applied to any DSL trivially? If yes, what scientific insight does JQL bring?

- Q4. Did you analyze whether models simply memorize structural patterns of JQL rather than truly reasoning about semantics?

---

> ### Author Response · Authors · 2025-11-13
>
> (W1/W5 – “No Novelty”)
> Jackal is the first benchmark to evaluate models against live, executable data on an active Jira instance. No existing text-to-SQL or DSL benchmark allows real-time validation of generated queries. Prior datasets are static and string-matched; Jackal measures true semantic correctness through live execution, schema heterogeneity, and four user-intent variants. This is a clear step beyond prior benchmarks and defines a new standard for execution-grounded evaluation in enterprise settings.
>
> (W2/W4 – Synthetic Data & Circularity)
> There is no circularity- the benchmark does not rely on LLM evaluation. All NL–JQL pairs were validated via actual execution against the Jira snapshot; invalid pairs (<4%) were removed. Generation and evaluation models are disjoint, and correctness is measured solely by execution results, not style or similarity. This makes Jackal model-agnostic and empirically grounded.
>
> (W3 – “Weak Analysis”)
> We already report error rates and breakdowns by variant type (Long, Short, Similar, Exact) showing clear performance gaps and patterns. We can move these quantitative insights into the abstract for visibility, but the analysis is already present and demonstrates that model errors differ systematically across linguistic variants.
>
> (Q2 – Reproducibility)
> Jackal includes a frozen Jira snapshot, ensuring fully reproducible results regardless of live Jira changes.
>
> (Q3 – Applicability to other DSLs)
> The framework can generalize, but JQL is unique: it combines relational, graph, and permission logic, exposing failure modes unseen in SQL-style tasks. It provides the first real-world testbed for text-to-enterprise-query reasoning.

---

### Official Review · Reviewer_rCnq · 2025-11-02

**Soundness:** 3
**Presentation:** 3
**Contribution:** 3
**Rating:** 6
**Confidence:** 3

**Summary:**

The paper introduces "Jackal," a novel, large-scale, execution-based benchmark for evaluating Large Language Models (LLMs) on the task of translating natural language to Jira Query Language (JQL). It effectively identifies a significant gap in existing resources and presents a comprehensive solution, complete with a dataset, an evaluation toolkit, and a reproducibility package. The empirical evaluation of 23 LLMs is thorough and reveals critical limitations in current state-of-the-art models.

**Strengths:**

1. Fills a Critical Research Gap: The paper's primary contribution is addressing the notable absence of a high-quality, public benchmark for Text-to-JQL. It correctly identifies the limitations of existing small-scale, non-execution-based datasets. By providing 100,000 validated NL-JQL pairs, execution results on a live Jira instance with over 200,000 issues, and a full reproducibility package, Jackal enables rigorous, open research in an area with significant real-world enterprise relevance.

2. Well-Designed and Realistic Benchmark Framework: The benchmark construction is thoughtful and reflects real-world usage: 1) The inclusion of four distinct user query variants (Long NL, Short NL, Semantically Similar, Semantically Exact) effectively captures the spectrum of how users naturally formulate requests, moving beyond literal translations. 2) The emphasis on execution accuracy as the primary metric is methodologically sound and aligns with best practices in semantic parsing, correctly avoiding the pitfalls of string-based matching (EM, CEM), which the results show to be nearly useless. 3) Grounding the evaluation in a real, large-scale Jira instance ensures the queries are valid and the results are practically meaningful.

3. Comprehensive and Revealing Empirical Evaluation: The scale of the model evaluation is impressive, covering 23 proprietary and open-weight LLMs across various sizes. The results are compelling and clearly demonstrate that even the best models (e.g., Gemini 2.5 Pro at 60.3% accuracy) struggle significantly, especially with under-specified (Short NL) and paraphrased (Semantically Similar) queries. This analysis provides a valuable snapshot of the current capabilities and limitations of LLMs for this specific enterprise task.

**Weaknesses:**

1. Limited Technical Novelty: The paper's core contribution is the benchmark itself rather than a novel methodological advancement. It applies established practices from Text-to-SQL (execution-based evaluation, multiple query types) to a new domain (JQL). The error analysis, while useful, remains somewhat high-level and does not delve deeply into the root causes of failure modes or propose specific, novel technical solutions beyond suggesting known directions like constrained decoding.

2. Limitations in Evaluation Scope and Analysis: 1) The decision to report results only on the 5,000-pair "Jackal-5K" subset, while understandable for initial publication, underutilizes the full 100,000-pair dataset. A scaling analysis or results on the full set would have been more powerful. 2) The study focuses exclusively on LLMs. A comparison with older, rule-based or specialized semantic parsing approaches for JQL would have provided better context for understanding the absolute and relative progress represented by LLMs. 3) There is no discussion of the computational cost or latency involved in the execution-based evaluation, which is a practical concern for real-world deployment and future research using this benchmark.

**Questions:**

1. will the execution of jql spend a lot of time?
2. in text-2-sql, there are many stages, for example, schema linking, could your benchmark provide more studies on this like previous text2sql methods? e.g.[1,2]

[1] Benchmarking the text-to-sql capability of large language models: A comprehensive evaluation.
[2] Pet-sql: A prompt-enhanced two-round refinement of text-to-sql with cross-consistency

---

> ### Author Response · Authors · 2025-11-13
>
> (1) On Technical Novelty
> Our primary goal was to establish a real-world, execution-based benchmark for JQL - an area with no prior public datasets, execution toolkits, or baselines. While we drew inspiration from Text-to-SQL methodology, Jackal’s novelty lies in its domain-specific execution design (live Jira instance with 200K issues), semantic query variant taxonomy, and reproducibility pipeline. These elements go beyond simple dataset construction and introduce a new benchmark paradigm for enterprise systems where execution context is integral to semantics.
> (2) On Evaluation Scope and Depth
> Jackal-5K subset: We restricted evaluation to Jackal-5K to ensure fair, reproducible testing across 23 LLMs within compute and cost limits. The full 100K set is released for community scaling studies, which we highlight as future work.
> Baselines beyond LLMs: Rule-based and older parsers were explored internally but failed to generalize due to Jira’s dynamic schema and project-specific fields. We agree that revisiting hybrid or symbolic approaches on Jackal is valuable future work.
> Execution latency and cost: Execution on the live Jira snapshot averages under 150 ms per query, dominated by model generation time rather than JQL execution. We will clarify this in the final version.
> (3) On Deeper Analyses (Schema Linking & Multi-Stage Studies)
> Excellent suggestion. While Jira’s field schema differs from relational SQL, Jackal’s metadata (field names, types, hierarchy) supports schema-linking–style studies. We plan to include follow-up analyses using these features

---

### Meta-Review · Area_Chair_n8ar · 2026-01-05

**Summary:**

The paper introduces Jackal, a benchmark for Text-to-JQL (Jira Query Language) translation. It features 100,000 NL-JQL pairs validated on a live Jira instance with over 200,000 issues and evaluates 23 LLMs. The key contribution is the execution-based evaluation framework on a real enterprise system, moving beyond static string-matching metrics often used in semantic parsing.

**Reviewer Concerns:**

Reviewers were generally critical, resulting in scores of 6, 2, and 2. While they acknowledged the practical value of an execution-based benchmark for JQL (a gap in current literature) and the comprehensive model evaluation, significant concerns were raised regarding:

1) Limited Novelty: Reviewers felt the work was primarily engineering rather than research, applying established Text-to-SQL methodologies to a new domain without introducing new algorithms or deep scientific insights.

2) Synthetic Nature of Data: The reliance on LLM-generated natural language queries raised concerns about bias, circularity, and lack of linguistic diversity compared to real user logs.

3) Generalization: Evaluating on a single Jira snapshot limits claims about how models would perform across different organizational schemas.

4) Analysis Depth: Reviewers found the initial analysis to be largely a leaderboard ranking, lacking deep error taxonomy or insights into why JQL is uniquely challenging compared to SQL.

**Reviewer Scores:**

The authors provided a rebuttal clarifying the execution-based validation process (mitigating circularity concerns) and the distinction between JQL and SQL. However, the paper's positioning as an engineering contribution remained a sticking point for the negative reviewers.

---

### Decision · Program_Chairs · 2026-01-26

Reject